# Geographic Variation in Testicular Morphometrics, Androgen Receptor Expression and Anti-Müllerian Hormone Levels in the Intermediate Roundleaf Bats across Distinct Regions in Thailand

**DOI:** 10.3390/ani13203287

**Published:** 2023-10-21

**Authors:** Kongkiat Srisuwatanasagul, Saritvich Panyaboriban, Sunate Karapan, Manita Wittayarat, Sayamon Srisuwatanasagul

**Affiliations:** 1Department of Anatomy, Faculty of Veterinary Science, Chulalongkorn University, Bangkok 10330, Thailand; kongkiat.s@chula.ac.th; 2Faculty of Veterinary Science, Prince of Songkla University, Songkhla 90110, Thailand; saritvich.p@psu.ac.th; 3Hala-Bala Wildlife Research Station, Hala-Bala Wildlife Sanctuary, Waeng, Narathiwat 96160, Thailand; karapann@gmail.com

**Keywords:** *Hipposideros larvatus*, bat, morphometric data, testis, androgen receptor, anti-Müllerian hormone

## Abstract

**Simple Summary:**

The intermediate roundleaf bat, *Hipposideros larvatus*, plays an ecological role in controlling insect populations. Studying its reproductive patterns, specifically testicular morphology, is essential for its conservation, especially in areas where it is at risk. This study focused on examining the testes morphology and the expression of two proteins, AR and AMH, in male *H. larvatus* bats from various parts of Thailand and during different sampling periods. Results indicated seasonal breeding patterns with significant differences in testes morphometric data, AR and AMH expression depending on the location and sampling period. Notably, bats from Dong Phayayen and Chiang Dao showed higher testicular activity in June, as evidenced by greater AR protein expression. By contrast, decreased AR expression was associated with lower testicular morphometric data and increased AMH expression. This suggests that AR is linked to active testicular functions, while AMH indicates a less active or regressive phase in *H. larvatus* bat in Thailand.

**Abstract:**

The *hipposideros larvatus* (intermediate roundleaf bat) is one of the insectivorous bats which has an agro-ecological role as a controller of the insect population. The reproductive patterns of *H. larvatus* are intricately linked to its ecological role and survival. An understanding of the testicular morphology can contribute to conservation for this species particularly in areas where its populations might be declining or under threat. However, these bats may also be associated with zoonotic diseases which can have significant public health implications. The aims of the study were to examine the morphological data as well as the expression of the androgen receptor (AR) and anti-Müllerian hormone (AMH) in the male reproductive organs of *H. larvatus* from different areas of Thailand and at different sampling periods. Their testes were processed for histological investigation and immunohistochemistry for AR and AMH. The results showed differences among the various sampling areas and different sampling periods, which suggested seasonal breeding characteristics. The higher testicular morphometric data were observed in *H. larvatus* from the Dong Phayayen (DY) and Chiang Dao (CD) areas during June, while the size of seminiferous tubules decreased thereafter. High AR immunostaining was noticed when the testicular morphometric data were higher in DY bats during June. On the other hand, low AR was observed in bats during August and September, which was concomitant with the decreases in seminiferous tubule size and germinal epithelial height. The results suggest a potential correlation between AR immunostaining and the active phase of testicular functions in *H. larvatus* during June which may imply the involvement of AR with the enhancement of testicular activity. Conversely, the low expression of AR may contribute to the upregulation of AMH in the testes and may indicate lower testicular activity in *H. larvatus* in Thailand.

## 1. Introduction

The reproductive system of bat species is complex and most exhibit the characteristics of seasonal breeders [1]. The bat reproductive cycle shows diversities depending on morphophysiological as well as environmental factors. In the male bat, the breeding cycle depends on geography, rainfall and photoperiod [2]. As a seasonal breeder, the bat testes show differences between the active and regressive periods of functions. The regressive stage of the testis is characterised by the decrease in seminiferous epithelial height and the interstitial connective tissue area with the decrease in cytoplasmic volume of Leydig cells [3]. In addition, a study on *Myotis nigricans* demonstrated that only Sertoli cells and spermatogonia could be observed in the testis during the regressive period [4]. The breeding season from tropical areas is from August to November and differs slightly between different species, even in the same habitation. Furthermore, an earlier study has shown the peak of spermatogenesis, testicular and epididymis sizes during July, followed by mating in August [5]. On the other hand, testicular regression is observed in the *Macrotus waterhousii* bats during late September and the testes are involuted by early December [6]. Regarding the reproductive cycle, the proliferation and apoptosis of male germ cells also varies in different periods of the year [7]. As very limited data exist on the male bat reproductive organs among different species, together with the fact that the health alerts from bat zoonosis are always issued during the bat breeding season, data on their reproductive cycle should be of interest. Until now, few studies have been carried out on the family Hipposideridae and a study of the male reproductive morphology in *Hipposideros larvatus* has not yet been performed.

The family Hipposideridae refers to insectivorous bats composed of 90 species. Among these, *H. larvatus* stands as one member, holding the designation of a species of least concern with the Red List of Threatened Species [8]. Roundleaf bats can be found in the Indo-Malaysian and Asian area, including Thailand. Notably, Hipposideros bats bear substantial ecological significance as agro-ecosystem contributors, playing a pivotal role as insectivorous animals which effectively regulate and control insect populations [9]. Given the distinctive reproductive traits of the roundleaf bat, characterized by an annual yield of one to two offspring, coupled with the high potential of habitat disturbance, the entire bat population experiences adverse effects. Consequently, the importance of comprehending the intricacies of their reproductive morphology and the dynamics of active reproductive behaviours becomes evident. Such insights stand to yield valuable perspectives regarding the potential influencers of pathogen transmission, especially during the critical breeding season.

The testis is the main target for androgens, acting via androgen receptors (ARs) in specific cells of the seminiferous tubule. Androgen plays an important role in testicular development, especially in the maturation of Sertoli cells, since the number of these cells has a crucial role in supporting the number of developing germ cells [10]. An earlier study of the *Artibius lituratus* testis demonstrated that AR localisation is restricted mainly to the nuclei of somatic cells rather than in the germ cells [4]. In contrast, studies of rats showed the expression of AR in different spermatogenic cells [11]. Despite this discrepancy in AR expression, the role of ARs in gametogenesis is widely accepted [10]. Furthermore, the AR expression in reproductive tissues is season-, cell- and reproductive-stage-specific [12]. The high expression of ARs in the testis during the active period compared to the hibernating or regressive period has been demonstrated in different species [4]. Studies of AR expression in the bat testis have been performed in different species [3,13]; nevertheless, there have been no studies on the testicular morphohistology or the expression of hormonal receptors in the male *H. larvatus* bat.

The anti-Müllerian hormone (AMH) in the testes of mammals is known to act negatively in testicular steroidogenesis and Leydig cell differentiation. AMH is a member of the transforming growth factor-beta superfamily that contributes to the development of seminiferous tubules. The level of AMH is comparatively high in the prepubertal period in most mammals and decreases at puberty with the onset of spermatogenesis [14]. Therefore, AMH has been suggested as a candidate marker for incomplete or reduced spermatogenesis [15]. However, the study of AMH during the inactive or regressive periods of the seasonally breeding testis is rare. According to the testicular function, the observed correlation between AR and AMH is that AR expression in the testis accounts for the downregulation in AMH and consequently coincides with spermatogenesis [16,17].

In the present study, we aimed to examine the morphological data of the male reproductive organs, mainly the testis of *H. larvatus*, using samples obtained from different areas of Thailand and at different sampling periods. The study included the expression of AR and AMH in the testis in addition to testicular morphometric data in order to investigate the relationship between these proteins and the reproductive function in male *H. larvatus*.

## 2. Materials and Methods

### 2.1. Animals

The testes were procured from *H. larvatus* bats, sourced from various areas across Thailand, spanning the period from June to September. The acquisition of samples was conducted under approved protocol in accordance with the standard observational procedure outlined “The survey for bat population and Rabies virus investigation”. This initiative falls under the purview of the Wildlife Research Division, Wildlife Conservation Office, Department of National Parks, Thailand (project year 2019–2021). In order to optimize the utilization of animal resources, the left testicular specimens underwent subsequent investigation to study their histomorphology. The specimens were collected from (1) Dong-Phayayen, Khao-Yai wildlife research station, Nakhon Ratchasima province, north-eastern (DY, *n* = 15), (2) Id cave, Chiang-Dao district, Chiang Mai province, northern (CD, *n* = 5), in June; (3) Nangkwak cave (NK, *n* = 6), Kamphaengphet province, central, in July, (4) Khaoruenai (URN, *n* = 3), (5) Khaothamlad temple (KTL, *n* = 3), Chachoengsao province, eastern, in August, and (6) Prathun cave (PT, *n* = 10), Lansak district, Uthaithani province, lower northern, in September. The details of sample collection including the distance from Bangkok, the centre of Thailand, and the mean temperature of each sampling point are shown in Table 1.

### 2.2. Morphological Study

Immediately after necropsy, the reproductive organs were dissected for anatomical study. Thereafter, the organs were fixed in 4% formaldehyde, slightly cut and processed histologically. All slides were stained with Haematoxylin and Eosin. In addition, the tissues were cut 4 mm thick and processed further for immunohistochemistry.

### 2.3. Immunohistochemistry

Immunohistochemistry was performed to study the localisation of AR and AMH as previously described by the earlier studies in other species [18,19]. Briefly, the antigen retrieval process was carried out by heating sections in citric acid using a microwave oven at 750 W. The primary antibodies used were rabbit polyclonal anti-AR (clone N-20, Santa Cruz Biotech, Dallas, TX, USA, 1:100) and mouse monoclonal AMH (clone B-11, 1:100). IgG isotype control was applied to tissue sections prepared for negative control slides. The biotinylated secondary antibodies relevant to the primary antibodies were applied by incubation with horseradish peroxidase avidin–biotin complex (Vectastain^®^, Vector Laboratories Inc., Burlingame, CA, USA). In order to visualise the bound enzyme, chromogen 3′,3′-diaminobenzidine (ImmPACT™ DAB kit, Vector Laboratories Inc., Burlingame, CA, USA) was applied in the final steps. All slides were counterstained with Haematoxylin, mounted in glycerine/gelatine, scanned by digital slide scanner and analysed by image analysis software (Caseviewer 2.4, 3DHISTECH, Budapest, Hungary).

### 2.4. Digital Slide Scanning and Image Analyses

Both morphological and immunohistochemical slides were studied using a digital slide scanner (3DHISTECH, Budapest, Hungary). Briefly, whole-slide digitalisation was applied to each sample section to visualise the whole picture. After slide scanning, testicular morphometrics were evaluated with Caseviewer. One hundred sections of seminiferous tubules were chosen randomly and measured. The seminiferous tubule diameter and the height of the germinal epithelium were measured, and their means were calculated. For the interstitial area, the investigation was performed by evaluating the interstitial area per 0.9 mm^2^ random area of testicular tissue.

For immunohistochemical staining, the expressions of AR and AMH protein in the testicular tissues of all samples were investigated by CellQuant and reported as a H-score. The immunohistochemical staining of each protein was categorised into different positive staining levels (high, moderate and low) and negative staining was set as default value for all evaluations. The positive staining intensity levels were determined based on multiple stained slides to ensure accurate measurement of all positive levels. Once the calibration was finalized, the settings were saved to a file which was then applied to all subsequent slide evaluations. Specific calibration levels were established for each protein under investigation, including AMH and AR. In the final step, the expressions of AR and AMH in the digital sections reported as a H-score were calculated from the formula [20]:H-score = (% high × 3) + (% moderate × 2) + (% low × 1)

### 2.5. Statistical Analyses

All statistical analyses were carried out using the R programming language (version 4.1.2). The morphometric data and H-score were presented as mean ± standard error of the mean (SEM). The Shapiro–Wilk test was used to confirm that the data were distributed normally. The statistical model was used to evaluate fixed effect (location and season), and the Wilcoxon rank-sum test with Bonferroni correction or ANOVA followed by the Tukey–Kramer test were used to assess the differences in the evaluations in nonparametric and parametric data, respectively.

## 3. Results

### 3.1. Morphometric Data

In this present study, all the examined bat carcasses were adults, with an average body weight of 22.77 ± 2.21 g and an average forearm length of 63.91 ± 1.60 mm. The testicular morphometric data of *H. larvatus* testes from different locations and different sampling periods are shown in Table 2 and Table 3, respectively. In general, the bats in all groups showed diversities in seminiferous tubule epithelia, though the elongated spermatids and spermatozoa were observed only in bats from the DY area in June. In other groups of bats, the elongated spermatids and spermatozoa were rarely found in the seminiferous tubules (Figure 1). The greatest testicular diameter was found in the DY in June and the smallest in KTL and PT bats collected during August and September, respectively. The same pattern was observed for the germinal epithelial height: the highest germinal epithelium was found in DY and the lowest in KTL and PT. Regarding different periods, greater testicular diameter was observed during June and July compared to later in the year, while the greatest germinal epithelial height was observed in *H. larvatus* during June. On the other hand, the interstitial area was relatively high in groups with a low testicular diameter and germinal epithelial height.

### 3.2. Immunohistochemistry

Immunolocalization of AR and AMH is shown as a H-score, which was calculated from both the intensity and proportion of the positive staining (Figure 2 and Figure 3). Positive stainings for AR and AMH immunolocalization were observed in the different cell types of the seminiferous tubule germinal epithelia, as well as in Sertoli cells (Figure 1). Positive AR immunostaining was found mainly in the nuclei of Sertoli cells with some faint cytoplasmic staining in the different types of germ cells in the seminiferous tubules. In addition, immunostaining of AMH was observed exclusively in the cytoplasm of both somatic cells and germ cells of the seminiferous tubules. For interstitial Leydig cells, positive AMH staining was observed in the cytoplasm, although the staining intensity and the proportion of positive cells were low. No staining was observed in the negative control slides for both AR and AMH immunolocalization (Figure 1). The highest AR H-score was found in the DY area, while lower AR H-scores were found from URN, KTL and PT; the KTL area showed no positive AR staining. On the other hand, the highest AMH H-score was found in *H. larvatus* from the URN and KTL areas. The lowest H-score of AMH immunostaining in the *H. larvatus* testis was found in the groups from the CD and DY areas (Figure 2). When comparing different sampling periods, the highest AR H-score was found only in June, while it was almost negative in other sampling periods. In contrast to the results of AR immunostaining, a high AMH H-score was found during August and September, while the lowest AMH H-score was found in June when the AR H-score was highest compared to the other periods (Figure 3). In addition, negative correlation was observed between the AR H-score and AMH H-score (r = −0.38, *p* < 0.01) in the testes of *H. larvatus.*

## 4. Discussion

The present study reported morphometric data on the testes of *H. larvatus* collected from different locations in Thailand and at different periods. It was widely accepted that Hipposideridae bats are seasonal breeders which was confirmed by our results, which showed a diversity in testicular morphometric data at different periods of the year. During June, the greatest seminiferous tubule diameter and germinal epithelial height were observed, indicating more active testes compared to the other periods of the year. Furthermore, it may imply the most suitable breeding time of the year for *H. larvatus* in Thailand. The morphometric data on *H. larvatus* testis is in accordance with those of other bat species, which demonstrate the breeding season when the temperature is warm [21]. On the other hand, the germinal epithelial height and the testicular diameter were reduced during August, which is the period of the low temperature in Thailand (in 2020); thus, it may not be suitable for the reproductive functions of Hipposideros bats. In a study of the male reproductive cycle of *Myotis nigricans* in Brazil, two peaks of spermatogenesis were observed, during September and January, with regression periods in November and June [4]. These observations in various bat species indicated that the diversity of testicular function may depend not only on the season but also on their habitation. The reproductive data of different species of the Hipposideridae have been reported: in *H. caffer* from Nigeria, spermatogenesis began in March and May, although the seminiferous tubules showed no spermatozoa or enlargement [22], while the enlargement of the testes in *H. cyclops* in Uganda was found during December [23]. This indicated a reproductive variation in the Hipposideridae in different species and habitations as well as at different periods of the year from those shown in our present results. Regarding the *H. larvatus* in Thailand, an earlier study showed that the highest mating activity was found during July and August in the Uthaithani province, but there were no morphometric data of the male gonad in that study [24].

From the results of AR immunostaining, the present study showed the highest AR H-score in the DY during June, which is concomitant with the greatest seminiferous tubule diameter and germinal epithelial height. The diameter of the seminiferous tubule in *H. larvatus* indicated an active reproductive period during June; consequently, the testes started to enter a period of regression during July, which was completed in August and September, as shown by the decreases in testicular size and germinal epithelial height. From the example from AR knockout mice, the absence of ARs in Sertoli and Leydig cells has a negative effect on spermatogenesis. Furthermore, a lack of AR in smooth muscle and peritubular myoid cells can cause a decrease in sperm output, although spermatogenesis was maintained [25]. However, data about the expression of AR in the bat testis are rare; thus, it has to be compared with other species. The differences in the AR H-score from the present investigation confirmed the variation in testicular function depending not only on the location of their habitation but also on the time of year. In addition, the greater seminiferous tubule size and germinal epithelial height in DY *H. larvatus* during June may indicate a more active testicular function that was mediated through the expression of AR in the testes compared to other locations and sampling periods. Regarding active and regressive testes, our findings were similar to those of the earlier study on *Myotis nigricans* [4] and *Molossus molossus* bats from Brazil [3], where a high AR expression was associated with the recrudescence stage of the testis, while a low AR expression was observed in the period of testicular deactivation [4]. In contrast to our results, the distribution of AR in the testes of *Artibeus literatus* bats increased during the non-reproductive phase [12]. The explanation of these findings from that earlier study was that the peak spermatogenic activity caused a negative feedback mechanism in the expression of AR in the testis. The discrepancy between our results and this earlier study may be due to the species specificity in reproductive physiology. Furthermore, androgen may act independently from Ars by a non-genomic pathway [26]; therefore, low AR expression could be observed during the testicular active phase. Besides the non-genomic pathway, the expression of Ars in the regressive testis of *H. larvatus* may be under other regulatory mechanisms in addition to regulation from circulatory and testicular androgen.

AMH is a member of the Transforming Growth Factor-β family that is expressed in the testis of both foetal and adult mammals [27,28]. The level of AMH has been widely used for the determination of gonadal function in both males and females [29] as well as in pathological conditions [19,30]. In the adult, a high expression of AMH in atrophied and dysfunctional testes may resemble that of the immature testis [31]. However, a study of the seasonally breeding Iberian mole suggested that the regulation of AMH expression was different between adults and juveniles [32]. In addition, inhibition of AMH may be used as a marker for the function of androgen in the onset of meiosis in the testis [33]. When downregulation of AMH in the testis is absent, onset of meiosis may not occur, resulting in low testicular function. For this reason, a high level of AMH during August and September when the germinal epithelial height was low may reflect the lack of AMH inhibition in the bat testes during the regression period. 

According to the results from the present study, it may suggest that geography and timing appear to be significant factors influencing the reproductive physiology of *H. larvatus*, as evidenced by discernible variations in testicular morphometric data and the expression levels of AR and AMH within different regions and sampling periods. Specifically, the higher testicular morphometric data observed in bats from the DY and CD areas during June might reflect optimal environmental and nutritional conditions favouring reproductive activity in these regions at that period. Nonetheless, it is essential to recognize a limitation in our study, especially sample collection. The bat carcasses were obtained from the survey study project, and consequently, the feasibility of obtaining a uniformly balanced distribution of samples across various locations and time points might have been limited. Moreover, aside from the geographical impacts on bat testes morphology, the expression of AR and AMH in bat testes is also influenced by other factors such as age and reproductive status. In our current findings, even though all sampled bats were adults, the expression levels of AR and AMH varied across different sampling periods. This variation might indicate that the reproductive state of the bats, whether they are in an active reproductive phase or a regressive one, plays a role in the expression of these proteins.

Regarding pathological conditions, a negative correlation between AMH expression and gonadal weight and sperm motility has been observed in mice treated with cytotoxic drugs. The increase in testicular AMH expression can cause an elevation of apoptosis and a reduction in cell proliferation [30]. Regarding the seasonal breeders, a high expression level of AMH genes during the non-breeding season and a low expression level during the breeding season has been shown in male animals [34]. Similarly, a study on the quail testis demonstrated an upregulation of AMH when the testis was seasonally regressed [35]. Consistent with these findings, the results from our study showed a higher AMH level in the testis of *H. larvatus* during August and September, in contrast to a low AMH during June and July. These studies, together with our results, suggest that AMH may play a role in the testicular cycle in seasonal breeding animals, including bats. 

The study of AR and AMH in the testis revealed that the downregulation of AMH in Sertoli cells was mediated through AR [36]. The decrease in AMH expression during puberty was found to coincide with the increase in AR in the testes of several mammals [10], as well as being accompanied by spermatogenesis [16]. The results of our study have shown the negative correlation between AR and AMH immunostaining in the testis of *H. larvartus*. This may suggest the reproductive period of this species to be when the expression of AR was high while AMH immunostaining was low. On the other hand, the regression period in the testis was demonstrated by the high AMH H-score and low negative AR H-score. In addition, some seasonally breeding bat species demonstrated two peaks of active and regressive testis throughout the year. For that reason, additional investigations at different periods of the year should be considered in further studies.

## 5. Conclusions

In conclusion, this study demonstrated, for the first time, the association between morphometric data, AR and AMH expression in the testis of *H. larvatus*. The variation in morphometric data and the expression of AR and AMH in the testis across different habitats and at different times of the year indicated the presence of seasonal breeding characteristics in this species. The higher testicular morphometric data, higher AR and lower AMH expression in bats during June may suggest an active reproductive period, while the lower testicular morphometric data, lower AR and higher AMH expression during August and September may suggest a testicular regression period in *H. larvatus* bats in Thailand.

## Figures and Tables

**Figure 1 animals-13-03287-f001:**
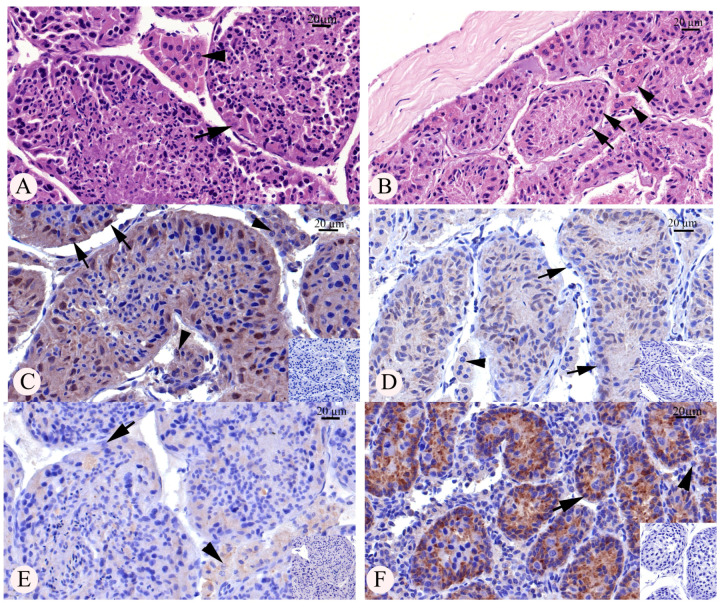
Histomorphology of *Hipposideros larvatus* bat testes (**A**,**B**) and immunolocalization of androgen receptor (**C**,**D**) and anti-Müllerian hormone (**E**,**F**) during active period (June; (**A**,**C**,**E**)) and regressive periods (August–September; (**B**,**D**,**F**)). Arrows represent Sertoli cells and arrow heads represent interstitial Leydig cells. AR immunostaining was found mainly in the nuclei of Sertoli cells (**C**), while strong AMH immunostaining was found in the cytoplasm of germ cells and Sertoli cells in the seminiferous tubule (**F**) with some faint staining in the interstitial Leydig cells (1E). Bars represent 20 μm and the insets in (**C**–**E**) show the negative control for each immunohistochemical staining.

**Figure 2 animals-13-03287-f002:**
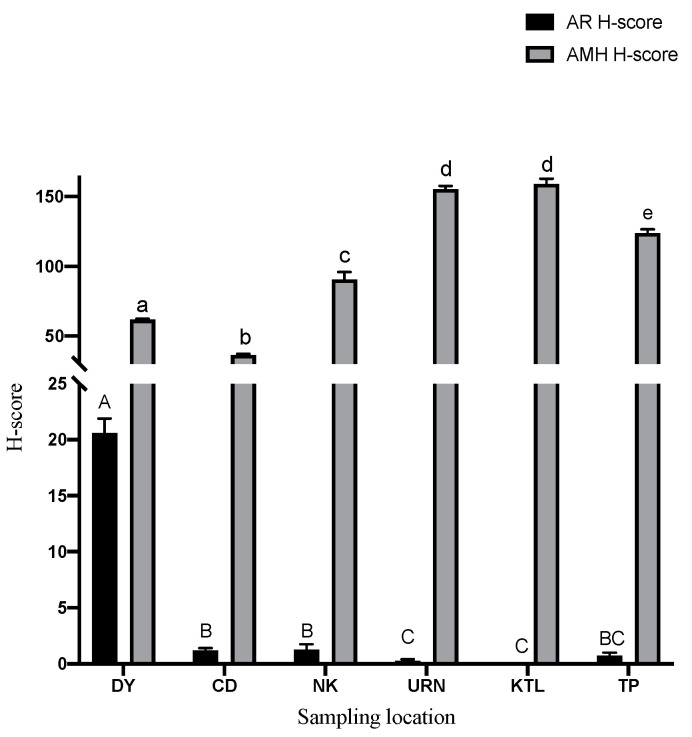
Immunohistochemical staining results in the *Hipposideros larvatus* bat testes from different sampling locations represented as H-score. Different letters in the same category indicate significant differences (*p* < 0.05).

**Figure 3 animals-13-03287-f003:**
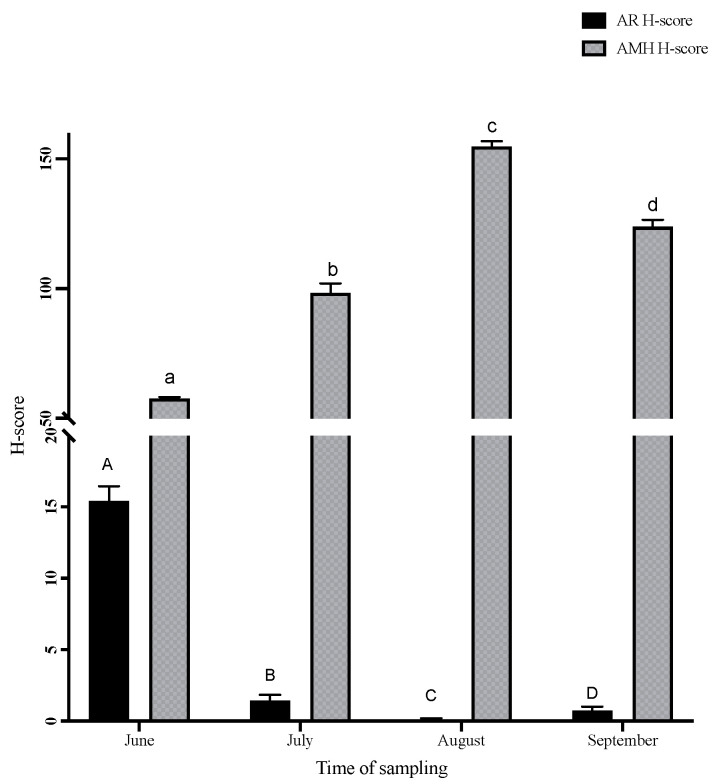
Immunohistochemical staining results in the *Hipposideros larvatus* bat testes from different sampling periods represented as H-score. Different letters in the same category indicate significant differences (*p* < 0.05).

**Table 1 animals-13-03287-t001:** The sample collections of *Hipposideros larvatus* bats from different locations and at different periods in Thailand.

Location/Mean Temperature	Sampling Period			
	June	July	August	September
Dong Phayayen (DY)/30.4 °C 160 km northeast of Bangkok	15	–	–	–
Chiang Dao (CD)/29.7 °C 772 km north of Bangkok	5	–	–	–
Nangkwak Cave (NK)/29.4 °C 390 km north of Bangkok	–	4	–	–
Khaoruenai (URN) 30 °C 167 km east of Bangkok	–	2	–	–
Khao Tham Lad Temple (KTL)/29.5 °C 162 km east of Bangkok	–	–	6	–
Prathun Cave (PT)/29.1 °C 201 km north of Bangkok	–	–	–	10

**Table 2 animals-13-03287-t002:** Morphometric data (mean ± SEM) in the *Hipposideros larvatus* bat testes from different sampling locations. Different letters in the same column indicate significant differences (*p* < 0.05).

Location	Seminiferous Tubule Diameter (µm)	Germinal Epithelium Height (μm)	Interstitial Area (µm²) per 0.09 mm² of Testicular Tissue (Area ± SEM) × 10^4^
1. DY (*n* = 15)	154.80 ± 0.38 ^A^	47.10 ± 0.15 ^A^	1.88 ± 0.026 ^A^
2. CD (*n* = 5)	127.44 ± 1.01 ^B^	34.92 ± 0.20 ^B^	2.18 ± 0.051 ^B^
3. NK (*n* = 3)	132.17 ±1.92 ^B^	22.35 ± 0.30 ^C^	2.62 ± 0.071 ^C^
4. URN (*n* = 3)	93.15 ± 1.35 ^BC^	20.77 ± 0.24 ^C^	2.80 ± 0.1 ^C^
5. KTL (*n* = 6)	72.76 ± 1.28 ^BC^	17.32 ± 0.17 ^D^	2.65 ± 0.10 ^C^
6. PT (*n* = 10)	79.89 ± 0.39 ^C^	18.38 ± 0.08 ^D^	3.09 ± 0.050 ^C^

**Table 3 animals-13-03287-t003:** Morphometric data in the *Hipposideros larvatus* bat testes from different sampling periods. Different letters in the same column indicate significant differences (*p* < 0.05).

Sampling Period	Seminiferous Tubule Diameter (µm)	Germinal Epithelium Height (µm)	Interstitial Area (µm²) per 0.09 mm² of Testicular Tissue (Area ± SEM) × 10^4^
June (*n* = 20)	145.91 ± 0.48 ^A^	43.01 ± 0.16 ^A^	1.96 ± 0.02 ^A^
July (*n* = 6)	135.83 ± 1.57 ^A^	23.00 ± 0.24 ^B^	2.56 ± 0.06 ^B^
August (*n* = 6)	86.39 ± 0.99 ^B^	19.39 ± 0.17 ^C^	2.73 ± 0.07 ^B^
September (*n* = 10)	79.89 ± 0.39 ^B^	18.38 ± 0.08 ^C^	3.09 ± 0.05 ^C^

## Data Availability

Data supporting the findings of this study are available from the corresponding author upon reasonable request.

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
