# Peer review of "Geographic Variation in Testicular Morphometrics, Androgen Receptor Expression and Anti-Müllerian Hormone Levels in the Intermediate Roundleaf Bats across Distinct Regions in Thailand"

_animals, 2023, doi:10.3390/ani13203287_

Round 1
Reviewer 1 Report
This paper was a very interesting and well presented article on the testicular morphometrics in roundleaf bats in Thailand. I think some clarification could be made to improve the paper. There are a few issues that I would like to see either resolved or explained further.
My biggest concern with the project is that throughout there is reference to different seasons and locations. However, because the samples were not collected equally from each location across the seasons it is hard to make a clear determination about the results being do to season or do to location or a combination of the two. I also think most readers will be unfamiliar with the geography of Thailand so more details regarding each location would be helpful. Approximately, how far away are they from each other? Is there variation in temperature and humidity? Are some more urban and therefore more likely to have light pollution? These are all factors that could be relevant but were not discussed. I think there needs to be clarification about if the goal was to characterize based on location or season as I don't see how both can be evaluated based on the sample collection strategy.
Page 3 lines 107-108. I believe this statement is implying that the samples were used for 2 separate studies. This sentence is poorly written and is confusing as it references studying testicular behavior. Testicles don't "behave" they may function a particular way. Was this to study the bat behavior relative to testicular analysis?
Page 3 Line 138 There must be a typo "Hundredsections" needs to be divided into two words but the sentence would still need improvement.
Page 3 Lines 143-148 This describes the categorization of staining levels. The criteria for where the cutoffs from one level to another were not described and needs to be included.
Results- Figures 1 and 2 are placed at the very end of the paper after Figure 3 with no mention of Figs 1&2 in the text. This needs corrected. I also think that Figure 3 may be presented as separate figures. I think the different traits that were being assessed could be presented as separate figures which would be easier to evaluate. Also the letters on the images are very hard to see. If the Figures were broken up, maybe they could be place just slightly out of the image. What objective/magnification was used for images?
In the text since all the data was presented in one figure, you need to include the letters that correspond to that particular analysis.
Discussion
Page 6 line 236. The use of "great effect" needs changed. In english great can mean large or positive. I think another word could clarify what the authors are trying to say.
Lines 248-250. This sentence describes how another study was in contrast to this one; however, it seems that different end points were measured. For example, one of the main points of the sentence was that AR was not correlated to testosterone. The present study did not measure testosterone so we really don't know if it was in contrast or not.
The english was very good. I have listed above where some minor changes could be made but overall, it was well done.
Reviewer 2 Report
This interesting study reports the geographic variation in testicular morphometrics, androgen re-2 ceptor expression and anti-Müllerian hormone levels in the in-3 termediate roundleaf bats across distinct regions in Thailand. Despite well written in general, there are some important points that should be adjusted before a final decision, as listed below:
Abstract - It is well written and clearly brings the main results obtained. However, the initial sentences that introduce the abstract only report the ecological importance of the given bats. Authors should also try to link that information to justify the importance of the present study for the species.
Introduction - Very well written. It clearly presents a background for the study. I only suggest authors to include the conservation status for the studied species, according to local offices and IUCN.
Material and Methods :
- Regarding the animals, how were they captured and euthanatized? Was there any criteria for the animals to be included in the study? For example, age, size, weight. Do you have any specific data related to this? If yes, please include it.
- Also, information related to reproductive status of the animals would be welcome, since the testis from prepubertal individuals is largely different from those of adult ones. At this sense, age would interfere in the data grouped per season. Please, discuss this.
- There are missing literature references for all the methods used.
- In statistics analysis, it is not clear what variables are being tested. What are you comparing? Seasons? Locations? Please, clarify.
Results
- The position of the tables at the end is not the best option. Table 01 should come at the methodology, others should come in results section.
- If animals were captured for monitoring some zoonotic diseases, any of them were infected? If yes, infections could impair testicular morphology or function. So, define if all the animals used were proven healthy.
- Even if results are clear, we have some doubt if the bats formed an homogeneous group in relation to their age and reproductive status. This could markedly impair the results. Authors should state if they were all adult. Did you find seminiferous tubule lumen in all the sample? All the animals presented clear signs of maturity?
Discussion
- As mentioned before, discussions should cover the influence of age and reproductive status of the animals besides season and location. Please consider revising it.
Round 2
Reviewer 1 Report
The authors did a great job responding to feedback. I understand it may be unavoidable when working with wildlife to collect samples perfectly. However, I still think there could be more discussion about geography vs timing.
Specific comments:
Page 4 Line 153 should be "One hundred sections" I should have caught that on my first review.
Page 4 Line 186 Put a space between Figure and 3.
Reviewer 2 Report
Authors conducted a very good revision on the manuscript. They addressed all my previous suggestions. Manuscript is now able to be accepted at the present form.
